# Association of social isolation, loneliness and genetic risk with incidence of dementia: UK Biobank Cohort Study

Marko Elovainio [1,2,3] Jari Lahti,[1] Matti Pirinen,[4,5,6] Laura Pulkki-Råback,[1,7] Anni Malmberg,[1] Jari Lipsanen,[1] Marianna Virtanen [8] Mika Kivimäki,[4,9] Christian Hakulinen[1,3]

For numbered affiliations see end of article.

**Correspondence to**
Dr Marko Elovainio;
marko.elovainio@helsinki.fi

## ABSTRACT

**Background** Social isolation and loneliness have been associated with increased risk of dementia, but it is not known whether this risk is modified or confounded by genetic risk of dementia.

**Methods** We used the prospective UK Biobank study with 155 070 participants (mean age 64.1 years), including self-reported social isolation and loneliness. Genetic risk was indicated using the polygenic risk score for Alzheimer's disease and the incident dementia ascertained using electronic health records.

**Results** Overall, 8.6% of participants reported that they were socially isolated and 5.5% were lonely. During a mean follow-up of 8.8 years (1.36 million person years), 1444 (0.9% of the total sample) were diagnosed with dementia. Social isolation, but not loneliness, was associated with increased risk of dementia (HR 1.62, 95% CI 1.38 to 1.90). There were no interaction effects between genetic risk and social isolation or between genetic risk and loneliness predicting incident dementia. Of the participants who were socially isolated and had high genetic risk, 4.4% (95% CI 3.4% to 5.5%) were estimated to developed dementia compared with 2.9% (95% CI 2.6% to 3.2%) of those who were not socially isolated but had high genetic risk. Comparable differences were also in those with intermediate and low genetic risk levels.

**Conclusions** Socially isolated individuals are at increased risk of dementia at all levels of genetic risk.

## INTRODUCTION

The rapidly rising numbers of people with dementia[1] is a significant health policy and health service concern in many high-income countries. Although considerable share of the dementia risk is due to genetic factors,[2 3] major efforts have been directed towards the identification of potentially modifiable risk factors that could prevent or delay the onset of dementia.[4] Higher levels of social support have been suggested to protect from dementia,[5] with both social isolation and feelings of loneliness being associated with increased risk of dementia,[6–8] although

### Strengths and limitations of the study

► The strengths of the study were its large sample size and a genome-wide study using a well-established polygenic risk score for dementia.
► Despite the large sample size, the sample was not representative of the UK population.
► As dementia was derived from hospital records, people with non-diagnosed dementia may have been missed.
► Reverse causation may have affected the findings by making people with preclinical dementia more socially isolated.
► Future research should examine the mechanistic pathways whereby social isolation is associated with dementia.

mixed findings have been reported between loneliness and dementia risk.[9 10] However, it remains unclear whether there is an interplay between genetic factors and social isolation and loneliness (ie, whether the association of social isolation and loneliness with dementia is evident only at high or low levels of genetic risk) or whether the associations of genetic factors and social network characteristics with dementia are independent and additive.

The polygenic risk score (PRS) for Alzheimer's disease (AD), describing the polygenic burden captured by the most recent genome-wide studies,[11] allows to estimate the size of the genetic risk and the extent to which the associations of social isolation and loneliness with dementia are modified by genetic risk. Existing studies have included Apolipoprotein E (APOE) genotype as the genetic risk, focused on wider psychosocial characteristics,[12] relied on small samples,[13] and provided limited evidence for the interplay of genetic risk and social relations predicting the increased risk of incident dementia. In the

present study, we used data from UK Biobank study to examine whether genetic risk may intensify and attenuate the associations of social isolation and loneliness with the risk of dementia. In addition to estimating relative risk, we will provide estimates of absolute risk,[14] as they are important information for risk communication and clinical risk prediction.[15]

## METHODS
### Study design and participants
In this analysis of the UK Biobank study, we used baseline data and obtained information of incident dementia at follow-up via linked electronic health records.[16] UK National Health Service (NHS) registers maintain records of all individuals legally registered as residents in the UK. In the UK Biobank study, these records were used to invite around 9.2 million individuals aged 40–69 years living within a sensible travelling distance of the 22 assessment centres across Great Britain 2007–2010.[16] At the study baseline, participants completed multiple touchscreen computer-based questionnaires followed by a face-to-face interview with trained research staff. Physical measures were also taken. Details of these assessments and variables are publicly available from the UK Biobank website: https://biobank.ndph.ox.ac.uk/ukb/.

In total, 502 656 individuals were recruited (5.4% of the eligible population). Of those, individuals that were 60 years or older and had complete data on social isolation, loneliness, dementia and genetic data were included in the present analysis (N=147 614–152 070). There were 7459 (4.8%) missing values in loneliness measures and 2351 (1.5%) missing values in isolation measures. We also repeated the analyses using imputed data in those with missing information on social isolation, loneliness or other explanatory variables but had information on genetic risk score (N=155 063).

### Ascertainment of incident dementia
Dementia was ascertained using hospital inpatient records which contains data on admissions and diagnoses from the Hospital Episode Statistics for England, Scottish Morbidity Record data for Scotland, and the Patient Episode Database for Wales. Additional cases were detected through linkage to death register data provided by the NHS Digital for England and Wales and the Information and Statistics Division for Scotland. Diagnoses were recorded using the International Classification of Diseases (ICD) coding system. Participants with dementia were identified as having a primary/secondary diagnosis (hospital records) or underlying/contributory cause of death (death register) using ICD-9 and ICD-10 codes for AD and other dementia classifications (see online supplemental file for details).

### Measurement of social isolation and loneliness
Social isolation and loneliness were measured using the same scale as in our two previous UK Biobank studies.[17 18]

*Social isolation* scale was defined using the following three questions: (1) 'Including yourself, how many people are living together in your household? Include those who usually live in the house such as students living away from home during term, partners in the armed forces or professions such as pilots" (one point for living alone); (2) 'How often do you visit friends or family or have them visit you?' (one point for friends and family visits less than once a month), and (3) 'Which of the following [leisure/ social activities] do you attend once a week or more often? You can select more than one', (one point for no participation in social activities at least weekly). This resulted in scale with a range from 0 to 3, where an individual was defined as socially isolated if he/she had two or more of those points and those who scored 0 or 1 were classified as not isolated. Other studies in the UK have used similar measures.[18]

*Loneliness* scale was constructed from two questions: 'Do you often feel lonely?' (no=0, yes=1) and 'How often are you able to confide in someone close to you?' (0=almost daily—once every few months 1=once every few months to never or almost never). An individual was defined as lonely if he/she responded positively to both questions (score 2) and not lonely if he or she responded negatively to one or both of the questions (score 0–1). Similar questions have been used in longer loneliness scales, such as the Revised UCLA Loneliness Scale.[19]

### PRS of dementia
From the genotyped UK Biobank samples, we included 155 070 unrelated white British participants after removal of participants based on heterozygosity and missingness of outliers, sex chromosome aneuploidies and mismatches, withdrawals and those that UK Biobank had excluded from the relatedness calculations. The genotypes were imputed against Haplotype Reference Consortium and UK10K haplotype resources containing ~96M variants.[11] We calculated PRS for AD based on a genome-wide association study by Kunkle *et al*[2] with 35 274 AD cases and 59 163 controls that do not overlap with UK Biobank samples (for details. see the online supplemental file). We used Plink 1.9[20] for the genotype QC and clumping. The following parameters were used for the clumping of the genotype data: p-value threshold 0.5, LD threshold ($r^2$) 0.5, and clumping window width of 250 kilobases. Prior to clumping, we excluded all single-nucleotide polymorphisms (SNPs) with minor allele frequency (MAF) <0.001, genotyping rate <0.1, Hardy-Weinberg equilibrium p-value<1e-6 and missingness per person >0.1. We used PRSice 2.2.8[21] for calculating the PRS with the genotype QC settings that have been recommended by the software developers.[22] In the main analyses, we applied a p-value threshold of 0.5, which resulted in including 626 623 SNPs in the PRS. This threshold was chosen as previous work has reported that it provided an optimal set of variants for predicting dementia and AD.[23 24] While this set is likely to include a number of variants which are not associated with AD, it also includes a number of variants

that at present do not have sufficient statistical evidence to meet the criteria for being genome-wide significant (ie, p value<5×10$^{-8}$) but are expected to be associated in future larger studies. The univariate associations between genetic risks score with 10 different cut-off points and incident dementia is reported in the online supplemental SFigure 1. Last, based on two single nucleotide polymorphisms (rs7412 and rs429358), we additionally genotyped APOE (none, one, or, two ε4 alleles).

The PRSs were then z-standardised to have mean 0 and variance 1, and divided into tertiles and categorised as low-risk, intermediate-risk and high-risk tertiles.

### Assessment of potential explanatory factors

Following information was used in the current study: sex, age in years, socioeconomic factors (educational attainment and Townsend deprivation index, which is an area-level composite measure of deprivation based on unemployment, non-home ownership, non-car ownership, and household overcrowding), chronic diseases (diabetes, cardiovascular disease, cancer, and other long-standing illness, disability or infirmity), cigarette smoking (smoker (yes/no); ex-smoker (yes/no)), physical activity (moderate and vigorous physical activity), alcohol intake frequency (three or four times a week or more vs once or twice a week or less), and the frequency of depressed mood in the past 2 weeks (Patient Health Questionnaire[25]).

### Statistical analyses

Study participants were followed from the study baseline (2006–2010) for incident dementia until the date of first dementia diagnosis, death, or to the end of the follow-up, whichever came first. The associations of social isolation, loneliness and PRS with incident dementia were examined using Cox proportional hazard regression models where age was used as a time scale. Results from these analyses were reported as HRs (relative risk) and their 95% CIs and the models were adjusted for age, sex and 10 first principal components of genetic structure from UK Biobank to control for possible population stratification, and additionally for education, social deprivation index, having long-term illness, physical activity, smoking status, alcohol consumption and depressive symptoms. In these analyses, PRS was used both as a categorical and as a continuous variable. Additional adjustments were also made for APOE genotype. Cumulative incidence (absolute risk) of dementia associated with combined categories of social isolation, loneliness and genetic risk was estimated using competing-risk regression,[26 27] with death being treated as competing event.

For the sensitivity analyses, missing data on social isolation, loneliness and all explanatory factors were imputed using multiple imputation by chained equations to generate five imputed datasets. Imputation model included age, sex, social isolation, loneliness, all covariates, the Nelson-Aalen estimate of cumulative hazard, and survival status.[28] Cox proportional hazards models were fitted within each imputed dataset and combined using Rubin's rules.

P values were two sided with statistical significance set at less than 0.05. All analyses were performed using Stata (V.15.1) and R (V.4.2.1).

### Role of the funding source

The sponsors of the study had no role in study design, data collection, data analysis, data interpretation, or writing of the report. Elovainio and Hakulinen had full access to the data. Elovainio and Hakulinen take final responsibility for the decision to submit for publication.

### Patient involvement

These results are based on existing data. We were not involved in the recruitment of the participants. As far as we know, no patients were engaged in designing the present research question or the outcome measures. They were also not involved in developing plans for recruitment, design or implementation of the study, and were not asked to advise on interpretation or writing up of results. Results from UK Biobank are disseminated to study participants via the study website and social media outlets.

### RESULTS

Descriptive statistics of the study participants are shown in table 1. Genetic risk score data were available for 155 070 participants (51.9% women; mean age 64.1 years). Overall, 8.6% of participants (n=13 103) were classified as socially isolated and 5.5% were lonely (N=8102). Of those who reported themselves to be socially isolated, 14.3% were also lonely. During a total of 1.36 million person years (mean follow-up time 8.8 years), 1444 participants (0.9% of the total sample) were diagnosed with all-cause dementia.

As expected, a higher PRS for AD was associated with an increased risk of dementia. Using continuous PRS, the HR per 1 SD increase in the score was 1.27 (95% CI 1.21 to 1.34) in an analysis adjusted for age, sex and 10 principal components. The associations between genetic risk categories (low, intermediate and high) with incidence of dementia shown in table 2. In comparison to the participants in the low genetic risk category, the HR of incident dementia was 1.49 (95% CI 1.28 to 1.73) in participants with intermediate risk and 1.71 (95% CI 1.47 to 1.98) in those with high genetic risk in the fully adjusted model. There were no interaction effects between sex and intermediate genetic risk (p=0.15) or between sex and high genetic risk (p=0.20) predicting incident dementia (online supplemental Stable 1a).

Social isolation was associated with increased risk of dementia (HR adjusted for age and sex=1.62, 95% CI 1.39 to 1.90). The associations attenuated but remained statistically significant after adjusting for additional covariates including sociodemographics, health-related factors and genetic risk score and principal components (HR=1.34,

**Table 1** Baseline characteristics of participants according to diagnosed dementia at follow-up

| Variables | | Dementia | | P value |
|---|---|---|---|---|
| | | No | Yes | |
| Age at baseline | Mean (SD) | 64.1 (2.8) | 65.8 (2.7) | <0.001 |
| Sex | Female | 79 816 (52.0) | 631 (43.7) | <0.001 |
| | Male | 73 803 (48.0) | 813 (56.3) | |
| Education | Lower | 40 575 (26.7) | 536 (38.2) | <0.001 |
| | Intermediate | 71 838 (47.4) | 606 (43.2) | |
| | Higher | 39 304 (25.9) | 261 (18.6) | |
| Long-term illness | No | 57 734 (38.7) | 319 (23.3) | <0.001 |
| | Yes | 91 264 (61.3) | 1053 (76.7) | |
| Physical activity | Low | 45 961 (30.7) | 479 (34.9) | 0.001 |
| | High | 103 933 (69.3) | 893 (65.1) | |
| Current smoker | No | 140 640 (92.0) | 1281 (89.4) | <0.001 |
| | Yes | 12 264 (8.0) | 152 (10.6) | |
| Alcohol consumption | Lower | 81 237 (52.9) | 866 (60.1) | <0.001 |
| | Higher | 72 281 (47.1) | 575 (39.9) | |
| Depressive symptoms | Low | 121 502 (82.5) | 1014 (75.8) | <0.001 |
| | Low medium | 21 350 (14.5) | 245 (18.3) | |
| | High medium | 2788 (1.9) | 42 (3.1) | |
| | High | 1639 (1.1) | 37 (2.8) | |
| Townsend deprivation index | Mean (SD) | −1.7 (2.8) | −1.1 (3.3) | <0.001 |
| Socially isolated | No | 138 407 (91.5) | 1208 (87.3) | <0.001 |
| | Yes | 12 922 (8.5) | 175 (12.7) | |
| Feeling lonely | No | 138 250 (94.5) | 1253 (92.5) | 0.001 |
| | Yes | 7999 (5.5) | 102 (7.5) | |
| Genetic dementia risk | Low | 51 355 (33.4) | 333 (23.1) | <0.001 |
| | Intermediate | 51 171 (33.3) | 517 (35.8) | |
| | High | 51 093 (33.3) | 594 (41.1) | |
| Apolipoprotein E genotype | None | 113 994 (74.2) | 707 (49.0) | <0.001 |
| | One e4 allele | 36 103 (23.5) | 568 (39.3) | |
| | Two e4 alleles | 3522 (2.3) | 169 (11.7) | |

95% CI 1.12 to 1.60). Loneliness was also associated with higher risk of dementia (HR=1.47, 95% CI 1.20 to 1.80), but this association was lost when adjusted for sociodemographics, health-related factors, PRS and principal components (HR=1.03, 95% CI 0.81 to 1.30). Both social isolation (HR=1.58, 95% CI 1.34 to 1.86) and loneliness (HR=1.28, 95% CI 1.03 to 1.59) were associated with incident dementia when added simultaneously into the

**Table 2** Association between genetic risk and risk of incident dementia

| Predictor | Model 1 | | Model 2 | |
|---|---|---|---|---|
| | HR (95% CI) | P value | HR (95% CI) | P value |
| Intermediate genetic risk versus low | 1.56 (1.36 to 1.79) | **<0.001** | 1.49 (1.28 to 1.73) | **<0.001** |
| High genetic risk versus low | 1.79 (1.56 to 2.04) | **<0.001** | 1.71 (1.47 to 1.98) | **<0.001** |
| Observations | 155 063 | | 139 345 | |

Model 1. Adjusted for age and sex.
Model 2. Adjusted for age, sex, education, social deprivation, health behaviours, long-term illness, depressive symptoms and 10 principal components.
The values are HR and 95% CI.

**Table 3** Associations of loneliness and isolation with incident dementia

| Predictor | Model 1 HR (95% CI) | P value | Model 2 HR (95% CI) | P value | Model 3 HR (95% CI) | P value |
|---|---|---|---|---|---|---|
| Separate analyses | | | | | | |
| Lonely versus not lonely | 1.47 (1.20 to 1.80) | <0.001 | 1.03 (0.81 to 1.30) | 0.817 | 1.04 (0.82 to 1.32) | 0.752 |
| Isolated versus no isolated | 1.62 (1.39 to 1.90) | <0.001 | 1.34 (1.12 to 1.60) | 0.002 | 1.34 (1.12 to 1.60) | 0.002 |
| Combined analyses | | | | | | |
| Lonely versus not lonely | 1.28 (1.03 to 1.59) | 0.024 | 0.95 (0.74 to 1.22) | 0.689 | 0.96 (0.75 to 1.23) | 0.716 |
| Isolated versus no isolated | 1.58 (1.34 to 1.86) | <0.001 | 1.33 (1.11 to 1.60) | 0.002 | 1.33 (1.11 to 1.60) | 0.003 |
| Observations | 147 604/152 712 | | 133 885/1 37 894 | | 133 885/137 894 | |

Model 1. Adjusted for age and sex.
Model 2. Adjusted for age, sex, education, social deprivation, health behaviours, long-term illness, depressive symptoms, genetic risk and 10 principal components.
Model 3. Adjusted for age, sex, education, social deprivation, health behaviours, long-term illness, depressive symptoms and apolipoprotein E genotype.
The values are HRs and 95% CI.

model but only the association between social isolation and dementia was robust to adjusting for additional covariates (HR=1.33, 95 % CI 1.11 to 1.60). Adjusting the models for APOE produced similar associations (table 3). No interaction effects between sex and isolation (p=0.53) or between sex and loneliness (p=0.14) predicting incident dementia were found (online supplemental Stable 1b).

Although no significant interaction effects in the associations between social isolation and genetic risk categories (p values range 0.45–0.62) or loneliness and genetic risk categories (p values range 0.59–0.95) with incident dementia were found (online supplemental Stable 1c), we illustrated the interplay between genetic risk with social isolation and loneliness by presenting associations at all genetic risk levels adjusting for potential confounders

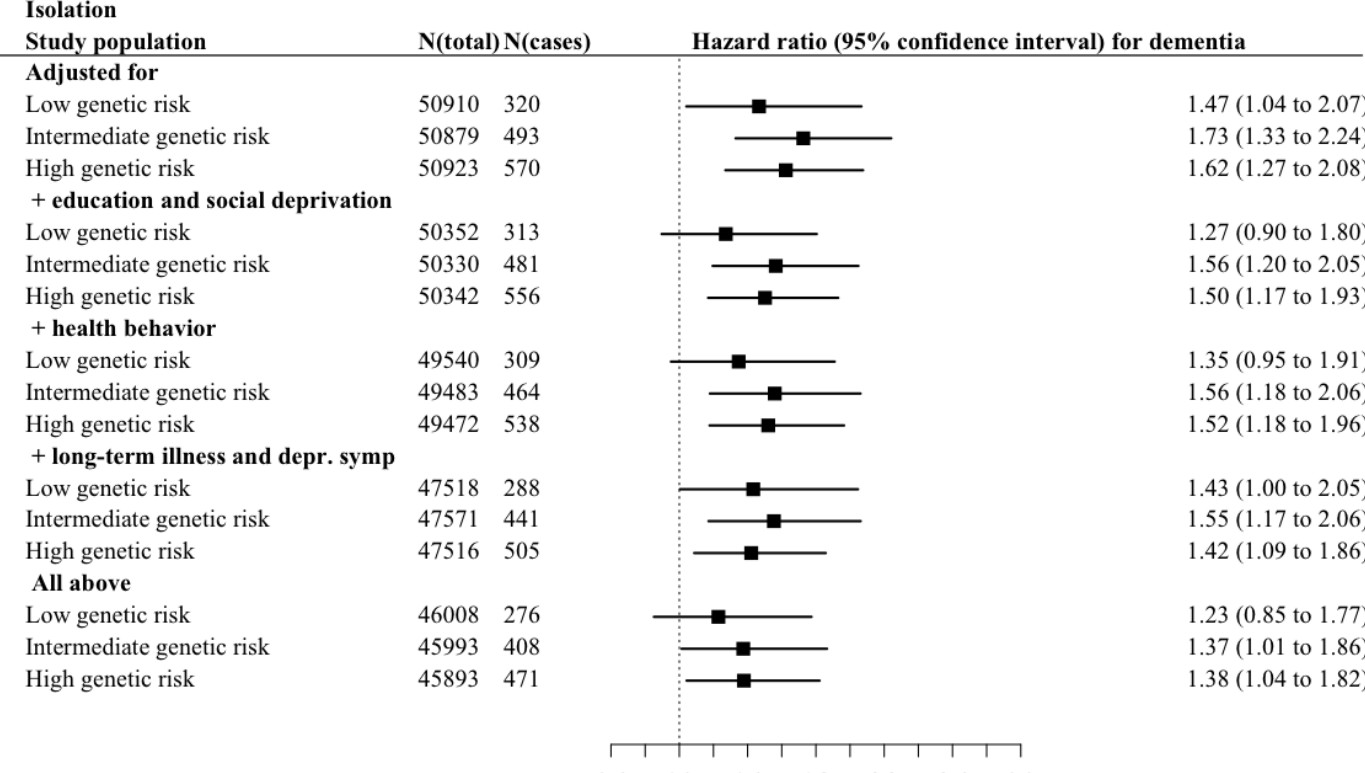

**Figure 1** Associations of social isolation with incident dementia risk in low, intermediate and high genetic risk groups.

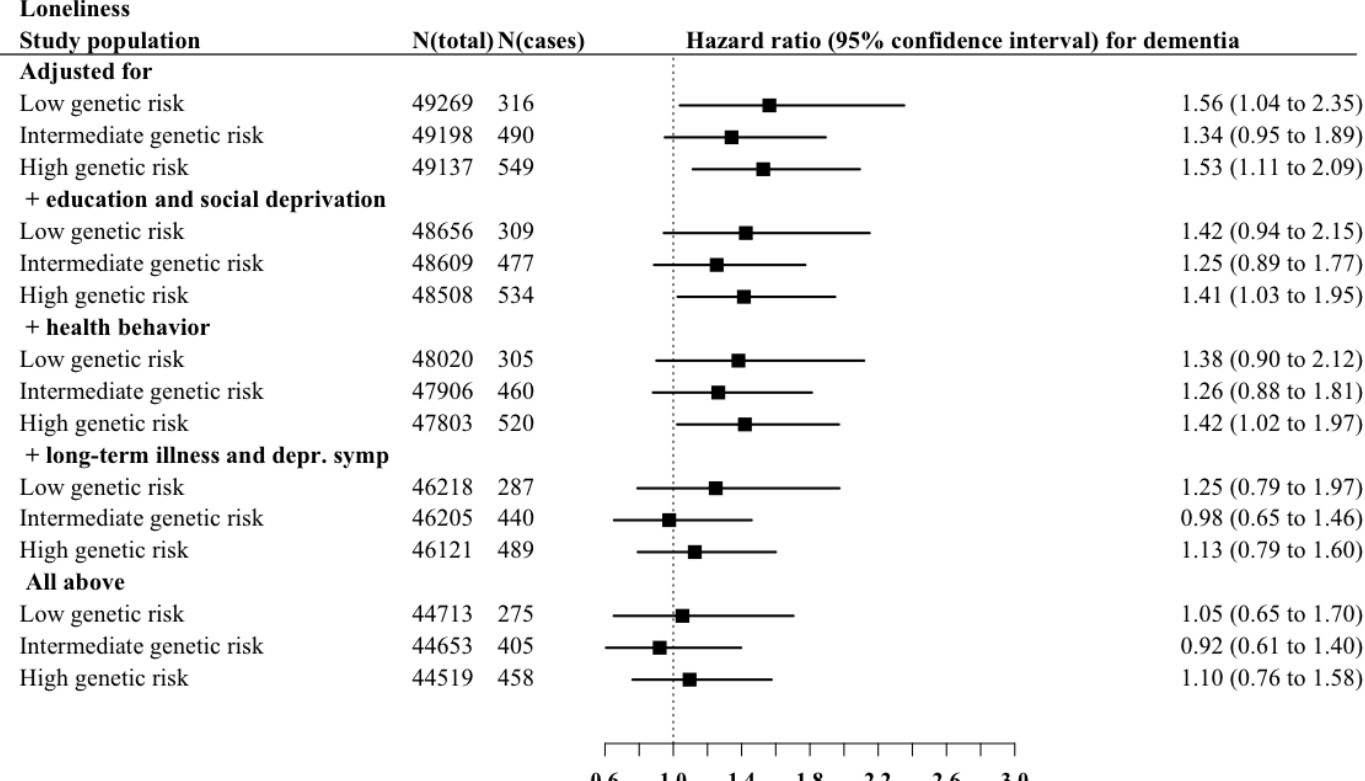

**Figure 2** Associations of loneliness with incident dementia risk risk in low, intermediate and high genetic risk groups.

(figures 1 and 2). Social isolation was associated with increasing dementia risk in all genetic risk levels. At intermediate and high genetic risk levels, these associations were robust to adjusting for all potential confounders or mediators (HR=1.37, 95% CI 1.01 to 1.86; HR=1.38, 95% CI 1.04 to 1.82). The results for loneliness were less consistent, and the risk of dementia was similar in lonely participants at low and at high levels of genetic risk, when compared with those who reported no loneliness. In the high genetic risk group, for example, the hazard ratios were 1.53 (95% CI 1.11 to 2.09) in low and 1.56 (95% CI 1.04 to 2.35) in high loneliness group (figure 2). All these association were attenuated when adjusted for long-term illness and depressive symptoms and in the fully adjusted model.

In terms of absolute risk (cumulative incidence), of those who were socially isolated and had high genetic risk, 4.4% (95% CI 3.4% to 5.5%) were estimated to developed dementia compared with 2.9% (95% CI 2.6% to 3.2%) of those who were not socially isolated but had high genetic risk (figure 3). The corresponding absolute risk estimates in the socially isolated and not isolated were 4.1 (95% CI 3.1% to 5.1%) and 2.5 (95% CI 2.2% to 2.8%) in participants with intermediate genetic risk and 2.3% (95% CI 1.5% to 3.0%) and 1.6% (95% CI 1.4% to 1.9%) in those with low genetic risk.

As sensitivity analyses, we repeated all the main analyses with AD as the outcome (online supplemental STables 2 and 3), and with missing explanatory variables imputed (online supplemental STable 4). The results did not

materially change. To detect whether the associations with incident dementia were due to reverse causation, we additionally repeated the fully adjusted models using data where those dementia cases occurring in the first 3 years of the follow-up were excluded. The association between isolation and incident dementia (HR=1.30, 95% CI 1.08 to 1.58) and between loneliness and incident dementia (HR=1.06, 95% CI 0.82 to 1.36) were similar.

## DISCUSSION

In this UK Biobank study of 155 063 men and women, social isolation was associated with increased risk of all-cause dementia and AD at intermediate and high levels of genetic risk of AD. No interaction effects were found between genetic risk levels and isolation predicting incident dementia. The incidence of dementia was estimated to reach over 4% in isolated high-genetic risk individuals compared with approximately 3% in non-isolated individuals with similar genetic risk. The difference between these groups was comparable also among those with intermediate and low genetic risk. This means that among individuals with similar genetic risk for dementia, those who are socially isolated are more likely to have incidence of the disease, suggesting an effect by social isolation over and above that of genetic risk. The association between loneliness and dementia was attributable to other dementia risk factors, such as health behaviours and depressive symptoms.

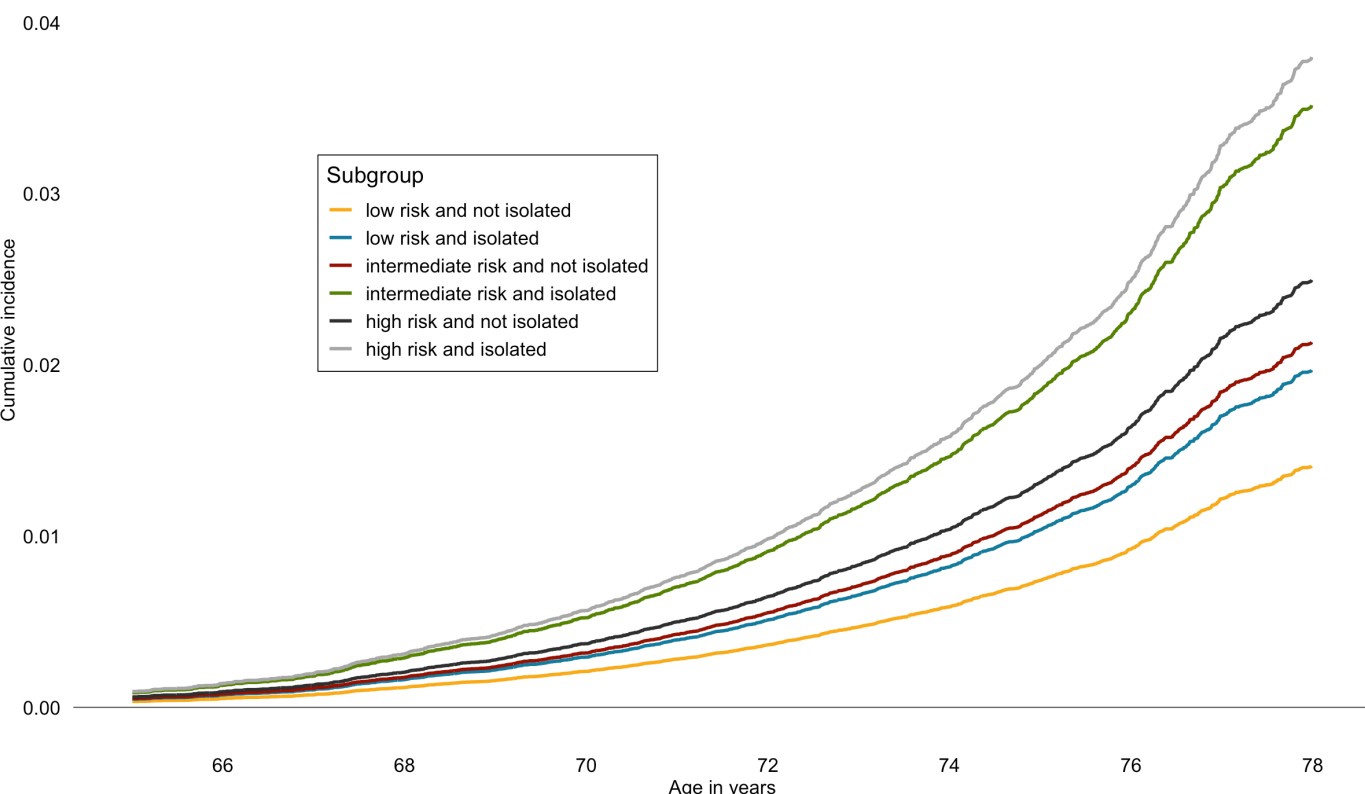

**Figure 3** Estimated cumulative incidence of dementia in combined genetic risk and social isolation groups.

The relative risk of dementia across the genetic risk categories was at the same magnitude as in a previous UK Biobank study[29] that used data from an older GWAS.[30] Our findings also support other studies—most of which with follow-ups from 5 to 11 years—showing an association of social isolation with increased risk of dementia.[6 8 10] A 28-year follow-up of 10 000 Whitehall II study participants found that less frequent social contacts at ages 50, 60 and 70 were associated with approximately 10% higher dementia risk, independent of socioeconomic and other lifestyle factors.[31] While previous studies have produced mixed findings on whether loneliness is associated with increased risk of dementia or not,[9 10] our findings show that the association between loneliness and dementia is mostly likely explained by other factors and present only at high levels of genetic risk.

Our results should be interpreted in a context of disease aetiology. Dementia is characterised by a 10–20 year preclinical or prodromal stage during which changes in biomarkers and cognitive abilities increasingly occur.[32] With a follow-up less than 10 years, it is likely that we assessed social isolation for dementia cases during this preclinical period. This could result to reverse causality, that is, increased prevalence of social isolation during the 8-year period could have resulted from preclinical changes in social activity leading to a spurious association between social isolation and dementia.

Several mechanisms through which social isolation may causally affect dementia risk have been proposed. Social isolation and loneliness have been suggested to increase stress reactivity which is associated with prolonged activation of the hypothalamic–pituitary–adrenal axis and the sympathoadrenal system.[33] This process may further lead to sleep deprivation, dysregulation of the immune system and even increased levels of oxidative stress,[34] all potentially harmful for cognitive health. It has also been shown that socially isolated and lonely individuals more often engage in health-damaging behaviours,[18] which may affect cognition either directly via biophysiological mechanisms or increased incidence of cardiometabolic diseases which accelerate neurodegeneration.[35] Socially isolated or lonely individuals are also at an increased risk of depression,[36] a potential risk factor for cognitive decline and dementia.[37] Participation in social activities and social interaction stimulates neural plasticity by building and maintaining cognitive reserve.[38] Poor cognitive reserve is a further pathway through which social isolation and loneliness could increase dementia risk.[39] Fewer social contacts with reduced exercising of memory and language adversely affect cognitive reserve, thereby accelerating dementia onset.[39] Cognitive ability was not assessed in the present study and a small share of the found association between social isolation and subsequent dementia risk may be attributable to lower initial cognitive reserve.

### Strengths and limitations

The major strengths of the current study include the large sample size of UK Biobank participants, which enabled us to study the combination of genetic risk, social isolation and loneliness in detail. In addition, we used the largest

genome-wide association study of dementia to date to derive the genetic risk for AD.[2]

There are also some important limitations. Although our analyses were adjusted for multiple potential sources of bias, the possibility of unmeasured confounding and reverse causation cannot be ruled out. However, the results were basically unchanged when excluding those with incident dementia during the first three-year follow-up time. Both frequency of social contacts and loneliness were self-reported and measured by relatively short and crude measures. As we were able to cover the genetic risk for AD—not all-cause dementias—based on the Kunkle *et al*,[2] we may have missed some of the genetic variance related to non-AD dementias. Dementia cases were derived from medical records or death registers, and thus some cases might have been missed. However, good agreement of dementia case determination with primary care record data has been shown.[40] This sample was restricted to volunteers of European ancestry aged 60–73 years at baseline and, therefore, further research is needed to ensure generalisability of our findings. As the mean age of participants was only 72 years at the end of the follow-up period, the incidence of dementia remained low. As noted previously, the response rate of the UK Biobank study survey was very low, 5.5%, and UK Biobank is not representative of the sampling population.[41] However, many etiological findings from UK Biobank appear to be generalisable to England and Scotland.[42]

## CONCLUSIONS

The present findings suggest an association between social isolation and increased risk of dementia across the spectrum of genetic risk. Further research is needed to determine the extent social isolation is a modifiable risk factor rather than a part of the dementia prodrome.

**Author affiliations**

[1]Department of Psychology and Logopedics, University of Helsinki, Helsinki, Finland
[2]Research Program Unit, Faculty of Medicine, University of Helsinki, Helsinki, Finland
[3]Finnish Institute for Health and Welfare, Helsinki, Finland
[4]Department of Public Health, University of Helsinki, Helsinki, Finland
[5]Institute for Molecular Medicine Finland, Helsinki Institute of Life Sciences, University of Helsinki, Helsinki, Finland
[6]Helsinki Institute for Information Technology and Department of Mathematics and Statistics, University of Helsinki, Helsinki, Finland
[7]Research Centre of Child Psychiatry, Faculty of Medicine, University of Turku, Turku, Finland
[8]School of Educational Sciences and Psychology, University of Eastern Finland, Joensuu, Finland
[9]Department of Epidemiology and Public Health, University College London, London, UK

**Acknowledgements** This research has been conducted using the UK Biobank Resource under Application Number 14801. Open access funded by Helsinki University Library.We thank the International Genomics of Alzheimer's Project (IGAP) for providing summary results data for these analyses. The investigators within IGAP contributed to the design and implementation of IGAP and/or provided data but did not participate in analysis or writing of this report. IGAP was made possible by the generous participation of the control subjects, the patients, and their families. The i–Select chips was funded by the French National Foundation on Alzheimer's disease and related disorders. EADI was supported by the LABEX (laboratory of excellence program investment for the future) DISTALZ grant, Inserm, Institut Pasteur de Lille, Université de Lille 2 and the Lille University Hospital. GERAD/PERADES was supported by the Medical Research Council (Grant no. 503480), Alzheimer's Research UK (Grant no. 503176), the Wellcome Trust (Grant no. 082604/2/07/Z) and German Federal Ministry of Education and Research (BMBF): Competence Network Dementia (CND) grant no. 01GI0102, 01GI0711, 01GI0420. CHARGE was partly supported by the NIH/NIA grant R01 AG033193 and the NIA AG081220 and AGES contract N01-AG-12100, the NHLBI grant R01 HL105756, the Icelandic Heart Association, and the Erasmus Medical Center and Erasmus University. ADGC was supported by the NIH/NIA grants: U01 AG032984, U24 AG021886, U01 AG016976, and the Alzheimer's Association grant ADGC-10-196728.

**Contributors** ME and CH designed the study and conducted the statistical analyses. ME wrote the first draft of the manuscript. JL and AM calculated the polygenetic risk score with the help of MP. All authors contributed to the interpretation of the results and critical revision of the manuscript for important intellectual content and approved the final version of the manuscript. ME and CH are the guarantors.

**Funding** ME and CH were supported by the Academy of Finland (339390 (ME)/310591(CH)). MK was supported by NordForsk (70521), the UK Medical Research Council (MRC S011676), the Academy of Finland (311492), and the US National Institutes on Ageing (NIA R01AG056477). LP-R was supported by the Jenny and Antti Wihuri Foundation. GERAD/PERADES was supported by the Medical Research Council (Grant no. 503480), Alzheimer's Research UK (Grant no. 503176), the Wellcome Trust (Grant no. 082604/2/07/Z) and German Federal Ministry of Education and Research (BMBF): Competence Network Dementia (CND) grant no 01GI0102, 01GI0711, 01GI0420. CHARGE was partly supported by the NIH/NIA grant R01 AG033193 and the NIA AG081220 and AGES contract N01-AG-12100, the NHLBI grant R01 HL105756, the Icelandic Heart Association, and the Erasmus Medical Center and Erasmus University. ADGC was supported by the NIH/NIA grants: U01 AG032984, U24 AG021886, U01 AG016976, and the Alzheimer's Association grant ADGC-10-196728.

**Competing interests** None declared.

**Patient consent for publication** Consent obtained directly from patient(s)

**Ethics approval** This study was conducted under generic approval from the NHS National Research Ethics Service (17 June 2011, Ref 11/NW/0382). Participants gave informed consent to participate in the study before taking part.

**Provenance and peer review** Not commissioned; externally peer reviewed.

**Data availability statement** Data are available upon reasonable request. The genetic and phenotypic UK Biobank data are available on application to the UK Biobank (www.ukbiobank.ac.uk). Summary statistics from the meta-analysis of genome wide association studies in dementia are available from https://www.niagads.org/datasets/ng00075.

**ORCID iDs**
Marko Elovainio http://orcid.org/0000-0002-1401-1910
Marianna Virtanen http://orcid.org/0000-0001-8361-3301

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
