## [Reviewer comments · BMJ Open]

ARTICLE DETAILS

TITLE (PROVISIONAL)	Association of social isolation, loneliness, and genetic risk with incidence of dementia: UK Biobank cohort study
AUTHORS	Elovainio, Marko; Lahti, Jari; Pirinen, Matti; Pulkki-Raback, Laura; Malmberg, Anni; Lipsanen, Jari; Virtanen, Marianna; Kivimaki, Mika; Hakulinen, Christian

VERSION 1 – REVIEW

REVIEWER	Holmans, Peter Cardiff University Institute of Psychological Medicine and Clinical Neurosciences
REVIEW RETURNED	27-Sep-2021

GENERAL COMMENTS	This manuscript studies the effects of social isolation and loneliness on dementia incidence in the UK Biobank sample. A novel and important feature of this study is the inclusion of genetic risk for Alzheimer's disease (modelled by polygenic risk score). The conclusions are clear, the manuscript is well written and the statistical methodology is appropriate. Major comments: 1. I would like to see the HRs for social isolation and loneliness presented separately for each stratum of genetic risk in addition to being shown in combination - this will indicate whether the effect of social isolation and/or loneliness depends on the level of genetic risk.2. More formally, the effect of the level of genetic risk on the HR associated with social isolation and/or loneliness could be tested by modelling genetic risk as a quantitative measure and including an interaction term of the PRS with social isolation (loneliness)3. Similarly, sex differences in the HR associated with social isolation (loneliness) can be tested by fitting interactions with sex.4. Given their large effects on dementia risk and onset, it would be good to model the effects of APOE genotypes separately from the PRS, if possible. If these are not available, they may be reconstructed from haplotypes of the relevant SNPs (rs429358 and rs7412) Minor comment 5. The point raised by the authors about reverse causation (social isolation actually being an early manifestation of dementia) is well
--

	taken. Including estimates of genetic risk for AD in the analyses may reduce this effect, even if it doesn't eliminate it. 6. Were the baseline measures of social isolation and loneliness used ?
--	--

REVIEWER	Dekhtyar, Serhiy Karolinska Institutet, Clinical Neuroscience – Psychology I have exchanged email with one of the authors of this study (Mika Kivimäki) about the possibility of collaborating in the future on an unrelated study on dementia risk. The work has not yet started and the only communication we've had was about the possibility of collaborating in the future. I described this situation to the BMJ Open editors and was instructed to proceed with the review, after disclosure.
REVIEW RETURNED	02-Dec-2021

GENERAL COMMENTS	Review BMJ Open – 2021-053936 Authors present a longitudinal study examining the interplay between polygenetic risk for Alzheimer's disease and social isolation/loneliness for the 8-year risk of dementia using UK Biobank data. They find that in addition to its main association with dementia risk, social isolation appears to accentuate the detrimental effects of genetic risk across all levels of PRS (although this conclusion is tentative, given the largely overlapping confidence intervals in the combination analysis). The findings for loneliness were less clear. While the paper is largely well written, and the analysis is mostly sound, authors fail to mention several prior studies have already examined the interplay between genetic risk and social network in relation to dementia. As such, the added value of these findings needs better justification. I also have concerns about potential reverse causation between social isolation and dementia, as well as the issue of cognitive impairment at the time of the baseline assessment. My detailed comments are below:  1. In the Background, authors write that the interaction between genetic factors and social isolation/loneliness in relation to dementia has not been much explored. This is not the case. For example, the interplay between APOE e4 status and social network has been assessed in the H70 cohort in Gothenburg, Sweden (PMID: 30587010). Another study from Sweden, based on the SNAC-K cohort, examined APOE-cognitive reserve (CR) interplay for dementia (and their measure of CR incorporated aspects of social connectedness and support; PMID: 31066941). Finally, a study from the Rotterdam cohorts, also used PRS together with an index of modifiable behaviors that also included social isolation (PMID: 31451782). Given the existence of these data, I would like the authors to better justify the need for their study, as well as clearly indicate their added value. 2. I am concerned about the possibility of the association between social isolation being due to reverse causation, whereby preclinical dementia symptoms lead to individuals disengaging from social participation. I was wondering if authors took any measures to try to limit its impact. One, could be to ensure the cognitively intact status of participants at baseline when the self-reports of social isolation are collected. Authors write nothing about cognitive screening at baseline; is it available/can it be used? Two, is to exclude dementia cases occurring in the first few years (3?) of the follow-up, with the hypothesis being that those who develop
---

	dementia early on are in the preclinical stage at baseline and therefore are likely subject to reverse causation. Could authors provide sensitivity analysis using only incident cases after some period of washout? 3. In the introduction, authors use the term social support as an overarching construct that appears to incorporate both social isolation and loneliness. In my experience, social support is generally viewed as a domain within social network that is meant to capture the qualitative aspects of social relations (vs. the extent of social connections that tackles the quantitative aspects of network). Authors may want to revise their usage of social support. 4. When describing the study population, authors write that they started out with just over 500,000 individuals, of whom those aged 60 years or more and with complete data on social isolation, loneliness, dementia status, and genetic risk were included, which constituted around 150,000 individuals. I would like to see a more detailed breakdown of the study population. How many were dropped because of missing data? A detailed study population flow chart would be extremely helpful. Those with missing data on loneliness, is there a way of understanding their characteristics with respect to other descriptives? Authors should provide a more careful synopsis of their study population as well as of those eligible yet excluded individuals. 5. What was the correlation and overlap between loneliness and social isolation? 6. Did the authors find any formal interaction between social isolation and genetic predisposition? In the absence of interaction, their interpretation of social isolation accentuating the impact of genetic predisposition may be too eager and ought to be tuned down. Instead, it may be more sensible to speak about the impact of social isolation being there across virtually all levels of genetic risk. 7. The writing in the abstract and in the discussion that describes the joint influence of social isolation and genetic risk is convoluted and very hard to follow. Please revise and simplify.
--	---

VERSION 1 – AUTHOR RESPONSE

#Reviewer: 1

Comments to the Author:

This manuscript studies the effects of social isolation and loneliness on dementia incidence in the UK Biobank sample. A novel and important feature of this study is the inclusion of genetic risk for Alzheimer's disease (modelled by polygenic risk score). The conclusions are clear, the manuscript is well written and the statistical methodology is appropriate.

We thank the reviewer for the positive feedback.

Comment 1: I would like to see the HRs for social isolation and loneliness presented separately for each stratum of genetic risk in addition to being shown in combination - this will indicate whether the effect of social isolation and/or loneliness depends on the level of genetic risk.

and

More formally, the effect of the level of genetic risk on the HR associated with social isolation and/or loneliness could be tested by modelling genetic risk as a quantitative measure and including an interaction term of the PRS with social isolation (loneliness)

Our response: We have now tested the interaction effects (a quantitative measure of genetic risk score * loneliness and a quantitative measure of genetic risk score * isolation) and reported the results in the additional supplement table (STable 1) and in the text as follows:

“Although no significant interaction effects in the associations between social isolation and genetic risk categories (p-values range 0.45-0.62) or loneliness and genetic risk categories (p-values range 0.59-0.95) with incident dementia were found (Stable 1c), we illustrated the interplay between genetic risk with social isolation and loneliness by presenting associations at all genetic risk levels adjusting for potential confounders (Figures 1 and 2).”

Comment 2: Similarly, sex differences in the HR associated with social isolation (loneliness) can be tested by fitting interactions with sex.

Our response: We tested the sex*loneliness and sex*isolation interactions and reported these in the text as follows: “No interaction effects between sex and isolation (p = 0.53) or between sex and loneliness (p = 0.14) predicting incident dementia were found (Stable 1b).”

Comment 3: Given their large effects on dementia risk and onset, it would be good to model the effects of APOE genotypes separately from the PRS, if possible. If these are not available, they may be reconstructed from haplotypes of the relevant SNPs (rs429358 and rs7412)

Our response: We additionally adjusted our models for the APOE genotypes. The text modifications in the Methods: “Last, based on two single nucleotide polymorphisms (rs7412 and rs429358), we additionally genotyped APOE (none, one, or, two ϵ 4 alleles).” and “Additional adjustments were also made for APOE genotype.” and in the Results: “Adjusting the models for APOE produced similar associations (Table 2).”

Comment 4: The point raised by the authors about reverse causation (social isolation actually being an early manifestation of dementia) is well taken. Including estimates of genetic risk for AD in the analyses may reduce this effect, even if it doesn't eliminate it.

Our response: Yes, we fully agree. We additionally repeated the analyses when excluding those with incident dementia during the first three follow-up years and reported the results as follows: “To detect whether the associations were due to reverse causation, we additionally repeated the fully adjusted models using data where those dementia cases occurring in the first three years of the follow-up were excluded. The association between isolation and incident dementia (hazard ratio= 1.30, 95% CI, 1.08-1.58) and between loneliness and incident dementia (hazard ratio= 1.06, 95% CI, 0.82-1.36) were basically the same.”

Comment 5: Were the baseline measures of social isolation and loneliness used ?

Our response: Yes, we used the baseline measures.

Reviewer: 2

Authors present a longitudinal study examining the interplay between polygenetic risk for Alzheimer's disease and social isolation/loneliness for the 8-year risk of dementia using UK Biobank data. They find that in addition to its main association with dementia risk, social isolation appears to accentuate the detrimental effects of genetic risk across all levels of PRS (although this conclusion is tentative, given the largely overlapping confidence intervals in the combination analysis). The findings for

loneliness were less clear. While the paper is largely well written, and the analysis is mostly sound, authors fail to mention several prior studies have already examined the interplay between genetic risk and social network in relation to dementia. As such, the added value of these findings needs better justification. I also have concerns about potential reverse causation between social isolation and dementia, as well as the issue of cognitive impairment at the time of the baseline assessment. My detailed comments are below:

Comment 1. In the Background, authors write that the interaction between genetic factors and social isolation/loneliness in relation to dementia has not been much explored. This is not the case. For example, the interplay between APOE e4 status and social network has been assessed in the H70 cohort in Gothenburg, Sweden (PMID: 30587010). Another study from Sweden, based on the SNAC-K cohort, examined APOE-cognitive reserve (CR) interplay for dementia (and their measure of CR incorporated aspects of social connectedness and support; PMID: 31066941). Finally, a study from the Rotterdam cohorts, also used PRS together with an index of modifiable behaviors that also included social isolation (PMID: 31451782). Given the existence of these data, I would like the authors to better justify the need for their study, as well as clearly indicate their added value.

Our response: We thank the reviewer for pointing us the missing literature. We have now included them and offered a further justification for our research as follows: "Existing studies have included APOE genotype as the genetic risk, focused on wider psychosocial characteristics [12], relied on small samples [13], and provided limited evidence for the interplay of genetic risk and social relations predicting the increased risk of incident dementia."

Comment 2: I am concerned about the possibility of the association between social isolation being due to reverse causation, whereby preclinical dementia symptoms lead to individuals disengaging from social participation. I was wondering if authors took any measures to try to limit its impact. One, could be to ensure the cognitively intact status of participants at baseline when the self-reports of social isolation are collected. Authors write nothing about cognitive screening at baseline; is it available/can it be used? Two, is to exclude dementia cases occurring in the first few years (3?) of the follow-up, with the hypothesis being that those who develop dementia early on are in the preclinical stage at baseline and therefore are likely subject to reverse causation. Could authors provide sensitivity analysis using only incident cases after some period of washout?

Our response: We have now repeated the main analyses without those that had incident dementia during the first 3 years of the follow-up and reported the results as follows: "To detect whether the associations were due to reverse causation, we additionally repeated the fully adjusted models using data where those dementia cases occurring in the first three years of the follow-up were excluded. The association between isolation and incident dementia (hazard ratio= 1.30, 95% CI, 1.08-1.58) and between loneliness and incident dementia (hazard ratio= 1.06, 95% CI, 0.82-1.36) were basically the same."

and in the discussion: "Although our analyses were adjusted for multiple potential sources of bias, the possibility of unmeasured confounding and reverse causation cannot be ruled out. However, the results were basically unchanged when excluding those with incident dementia during the first three - year follow-up time."

Comment 3. In the introduction, authors use the term social support as an overarching construct that appears to incorporate both social isolation and loneliness. In my experience, social support is generally viewed as a domain within social network that is meant to capture the qualitative aspects of social relations (vs. the extent of social connections that tackles the quantitative aspects of network). Authors may want to revise their usage of social support.

Our response: We have changed the text as follows: "However, it remains unclear whether there is an interplay between genetic factors and social isolation and loneliness (i.e. whether the association of

social isolation and loneliness with dementia is evident only at high or low levels of genetic risk) or whether the associations of genetic factors and social network characteristics with dementia are independent and additive.“

Comment 4. When describing the study population, authors write that they started out with just over 500,000 individuals, of whom those aged 60 years or more and with complete data on social isolation, loneliness, dementia status, and genetic risk were included, which constituted around 150,000 individuals. I would like to see are more detailed breakdown of the study population. How many were dropped because of missing data? A detailed study population flow chart would be extremely helpful. Those with missing data on loneliness, is there a way of understanding their characteristics with respect to other descriptives? Authors should provide a more careful synopsis of their study population as well as of those eligible yet excluded individuals.

Our response: We have now explained the number of missing values in loneliness and isolation variables in the Methods as follows: “There were 7459 (4.8%) missing values in loneliness measures and 2351 (1.5%) missing values in isolation measures” We also provided the results using the imputed data in the Supplement (Stable 4).

Comment 5. What was the correlation and overlap between loneliness and social isolation?

The association between loneliness and isolation is illustrated as a cross table and the result provided in the Results section as follows: “Of those who reported themselves to be socially isolated 14.3.% were also lonely.”

Comment 6. Did the authors find any formal interaction between social isolation and genetic predisposition? In the absence of interaction, their interpretation of social isolation accentuating the impact of genetic predisposition may be too eager and ought to be tuned down, Instead, it may be more sensible to speak about the impact of social isolation being there across virtually all levels of genetic risk.

Our response: We have now tested the interaction effects (a quantitative measure of genetic risk score * loneliness and a quantitative measure of genetic risk score * isolation) and reported the results in the additional supplement table (STable 1) and in the text as follows:

“Although no significant interaction effects in the associations between social isolation and genetic risk categories (p-values range 0.45-0.62) or loneliness and genetic risk categories (p-values range 0.59-0.95) with incident dementia were found (Stable 1c), we illustrated the interplay between genetic risk with social isolation and loneliness by presenting associations at all genetic risk levels adjusting for potential confounders (Figures 1 and 2).”

Comment 7. The writing in the abstract and in the discussion that describes the joint influence of social isolation and genetic risk is convoluted and very hard to follow. Please revise and simplify.

Our response: We have partly re-written the Abstract and the Discussion to simplify the message and reflect the new results.

VERSION 2 – REVIEW

REVIEWER	Holmans, Peter Cardiff University Institute of Psychological Medicine and Clinical Neurosciences
-----------------	--

REVIEW RETURNED	20-Jan-2022
GENERAL COMMENTS	The authors have done a good job of responding to the reviewers' comments, and the manuscript is much improved. I recommend acceptance subject to the following minor changes: To improve clarity, the first sentence of the final paragraph of the Results section (Page 12), could be rewritten as "As sensitivity analyses, we repeated all the main analyses with Alzheimer's disease as the outcome (STables 2-3), and with missing explanatory variables imputed (STable 4)". And the third sentence could start "To detect whether the associations with incident dementia were due to reverse causation..." (since the authors have just mentioned Alzheimer's disease as an outcome). Finally, the phrase "basically the same" doesn't sound very scientific - it could be replaced by "similar".
REVIEWER	Dekhtyar, Serhiy Karolinska Institutet, Clinical Neuroscience - Psychology I have provided my coi statement when submitting my first review. Briefly, I have exchanged email with one of the co-authors of this study (Mika Kivimäki) about the possibility of collaborating on an unrelated project. The work in question hasn't started yet, and we have no other ongoing collaborations/contact.
REVIEW RETURNED	14-Jan-2022
GENERAL COMMENTS	I thank the authors for addressing my feedback. I have no more comments.

VERSION 2 – AUTHOR RESPONSE

Reviewer: 1

The authors have done a good job of responding to the reviewers' comments, and the manuscript is much improved. I recommend acceptance subject to the following minor changes:

To improve clarity, the first sentence of the final paragraph of the Results section (Page 12), could be rewritten as "As sensitivity analyses, we repeated all the main analyses with Alzheimer's disease as the outcome (STables 2-3), and with missing explanatory variables imputed (STable 4)". And the third sentence could start "To detect whether the associations with incident dementia were due to reverse causation..." (since the authors have just mentioned Alzheimer's disease as an outcome). Finally, the phrase "basically the same" doesn't sound very scientific - it could be replaced by "similar".

Our response:

We thank the reviewer for the positive feedback and. All the changes have been done as suggested.